# The Effect of Cold-Water Swimming on Energy Metabolism, Dynamics, and Mitochondrial Biogenesis in the Muscles of Aging Rats

**DOI:** 10.3390/ijms25074055

**Published:** 2024-04-05

**Authors:** Mateusz Bosiacki, Maciej Tarnowski, Kamila Misiakiewicz-Has, Anna Lubkowska

**Affiliations:** 1Department of Biochemistry and Medical Chemistry, Pomeranian Medical University in Szczecin, Powstańców Wlkp. 72, 70-111 Szczecin, Poland; 2Department of Physiology in Health Sciences, Pomeranian Medical University in Szczecin, Żołnierska Str. 54, 71-210 Szczecin, Poland; maciejt@pum.edu.pl; 3Department of Histology and Embryology, Pomeranian Medical University in Szczecin, 72 Powstańców Wielkopolskich Str., 70-111 Szczecin, Poland; kamila.misiakiewicz.has@pum.edu.pl; 4Department of Functional Diagnostics and Physical Medicine, Faculty of Health Sciences, Pomeranian Medical University in Szczecin, Żołnierska Str. 54, 71-210 Szczecin, Poland; anna.lubkowska@pum.edu.pl

**Keywords:** cold-water swimming, aging, energy metabolism, mitochondrial dynamics, mitochondrial fusion and fission, peroxisome proliferator-activated receptor-γ coactivator (PGC-1α), mitofusin 1 (Mfn1), mitofusin (Mfn2), optic atrophy 1 protein (Opa1), dynamin-related protein 1 (Drp1)

## Abstract

Our study aimed to explore the potential positive effects of cold water exercise on mitochondrial biogenesis and muscle energy metabolism in aging rats. The study involved 32 male and 32 female rats aged 15 months, randomly assigned to control sedentary animals, animals training in cold water at 5 ± 2 °C, or animals training in water at thermal comfort temperature (36 ± 2 °C). The rats underwent swimming training for nine weeks, gradually increasing the duration of the sessions from 2 min to 4 min per day, five days a week. The results demonstrated that swimming in thermally comfortable water improved the energy metabolism of aging rat muscles (increased metabolic rates expressed as increased ATP, ADP concentration, TAN (total adenine nucleotide) and AEC (adenylate energy charge value)) and increased mRNA and protein expression of fusion regulatory proteins. Similarly, cold-water swimming improved muscle energy metabolism in aging rats, as shown by an increase in muscle energy metabolites and enhanced mitochondrial biogenesis and dynamics. It can be concluded that the additive effect of daily activity in cold water influenced both an increase in the rate of energy metabolism in the muscles of the studied animals and an intensification of mitochondrial biogenesis and dynamics (related to fusion and fragmentation processes). Daily activity in warm water also resulted in an increase in the rate of energy metabolism in muscles, but at the same time did not cause significant changes in mitochondrial dynamics.

## 1. Introduction

Despite the warnings from scientists and physicians that the modern sedentary, overfed, and overstimulated lifestyle contributes to the rise of civilization-related diseases, the understanding of this issue remains inadequate [1]. Moreover, there is a lack of comprehension regarding the potential role of simple lifestyle changes in preventing various diseases, particularly during the aging process [2,3]. With the advancements in civilization and technology, humans now experience less physical discomfort from environmental factors such as low temperatures or intense physical exertion. As a result, the body’s adaptive mechanisms are becoming less efficient, leading to a higher susceptibility to stress-inducing factors, both internal and external. This, in the long run, can deplete health resources and put a strain on the well-being of older individuals with reduced physiological adaptability [4,5].

One crucial aspect of “healthy aging” lies in physical activity and a balanced diet. Physical exertion, including cold water immersion (even in natural water bodies), is widely believed to have positive effects on both mental and physical health. However, the existing scientific literature provides limited evidence to support this belief. While physical activity in a cold environment is known to influence the energetic metabolism of various tissues, little data exists on its impact on mitochondrial biogenesis and energetic metabolism in muscles [6,7,8]. 

Mitochondria play a critical role in metabolism, serving as the site for numerous biochemical processes, such as oxidative phosphorylation, the Krebs cycle, beta-oxidation of fatty acids, calcium buffering, and heme biosynthesis. They are also involved in steroid and heme synthesis, contribute to the regulation of thermogenesis processes, and influence cell aging and death through apoptosis and necrosis. Thus, mitochondria are not solely responsible for energy production but also vital for signal transduction and cell survival [9]. Consequently, disruptions in mitochondrial function, particularly in their metabolic activity, are associated with various disorders, including metabolic and neurodegenerative diseases, cancers, and the aging process [10,11]. 

Mitochondria are not static structures but undergo recurring fusion and fission processes [12,13]. This phenomenon, known as mitochondrial dynamics, enables mitochondria to adapt their morphology to meet the metabolic needs of the entire cell.

Mitochondrial fusion serves a crucial purpose in facilitating the exchange of mtDNA between mitochondria, preventing the accumulation of mutated mitochondrial genes. This process also ensures an even distribution of proteins and metabolites throughout the mitochondrial network and allows for the incorporation of damaged mitochondria, offering the possibility of restoring their normal function. Any disruptions in the fusion process can lead to reduced inner mitochondrial membrane potential and impaired respiratory chain activity [14]. Mitochondrial fission, on the other hand, enables the transportation of mitochondria within the cell, ensuring their equitable distribution to daughter cells during division and playing a critical role in mitophagy [15].

In mammals, the mitofusin family of proteins (Mfn1 and Mfn2) mediates the fusion of the outer mitochondrial membrane, while the optic atrophy 1 protein (Opa1) participates in the fusion of the inner mitochondrial membrane. Mitochondrial fragmentation is facilitated, in part, by the protein dynamin-related protein 1 (Drp1) [14,15].

Another important player in cellular metabolism is the peroxisome proliferator-activated receptor-γ coactivator (PGC-1α), which collaborates with the PPARγ receptor to regulate the expression of various genes. PGC-1α is induced mainly during increased tissue oxygen demand, such as during heightened physical activity in skeletal muscles, and exposure to cold in brown adipose tissue. Its activation leads to an increase in mitochondrial numbers and the induction of gene expression related to fatty acid oxidation and processes like heme synthesis. PGC-1α interacts with other transcription factors, fulfilling crucial metabolic functions in cells, making it a central regulator of the body’s energy metabolism [16].

Considering the potential effects of cold exposure as a stressor, repeated cold water exercise may trigger adaptive responses, which could positively impact the aging process in muscles. Thus, it is reasonable to explore the hypothesis that physical exertion in cold water, tailored to individual considerations like temperature, duration, form of exposure, and gender, may enhance mechanisms involved in mitochondrial biogenesis and ATP (adenosine-5’-triphosphate) synthesis in aging rat muscles.

The objective of this study was to assess the influence of physical activity in cold water on the energy status, purine compound content, dynamics, and mitochondrial biogenesis in aging rat muscles. The detailed measurements included the following: the concentration of ATP; ADP (adenosine-5’-diphosphate); AMP (adenosine-5’-monophosphate); Ado (adenosine); the values of adenylate energy charge (AEC); the values of total adenine nucleotide (TAN = ATP + ADP + AMP); the expression of mRNA and the protein of PGC-1α; mitofusin 1 (Mfn1); mitofusin 2 (Mfn2); Opa1; Drp1 in the skeletal muscles of aging rats subjected to physical training in cold water.

## 2. Results

### 2.1. ATP Concentration in the Muscle of Rats

The concentration of ATP in the skeletal muscles of the animals showed significant variations among the groups. In male rats that swam in cold water (5 °C group), the ATP concentration was significantly higher compared to both the control group (by approximately 64%, *p* ≤ 0.001) and the 36 °C group subjected to warm water training (by approximately 30%, *p* ≤ 0.001). Similarly, in male rats that swam in warm water, ATP concentrations were higher (by 14%) compared to the control group (*p* ≤ 0.001) (Figure 1).

Likewise, female rats swimming in cold water exhibited higher ATP concentrations in their muscles (by approximately 33%, *p* ≤ 0.001) compared to the control group. The ATP concentration in the muscles of female rats undergoing warm water training was also higher (by approximately 15%, *p* ≤ 0.001) compared to the control group. Notably, the ATP concentration in the muscles of female rats swimming in cold water was significantly higher than those undergoing training in warm water (by approximately 16%, *p* ≤ 0.001) (Figure 1). It is worth mentioning the observed intersexual variation in the swimming groups, which was not observed in the control groups. ATP levels were higher in males compared to females in both the 5 °C group (*p* ≤ 0.001) and the 36 °C group (*p* = 0.002).

### 2.2. ADP Concentration in the Muscle of Rats 

Regarding ADP concentrations in the muscles of male rats, those swimming in cold water had significantly higher levels compared to both the control group (by approximately 21%, *p* ≤ 0.001) and the rats swimming in warm water (by approximately 12%, *p* = 0.003). In male rats swimming in warm water, ADP levels were also higher (by approximately 6%, *p* = 0.157) compared to the control group, but this difference was not statistically significant (Figure 1).

Similarly, female rats undergoing swimming training in cold water exhibited significantly higher ADP concentrations (by approximately 54%, *p* ≤ 0.001) compared to the control group. Female rats in the 36 °C group also showed higher ADP concentrations in their muscles (by approximately 30%, *p* ≤ 0.001) compared to the control group. Moreover, the ADP concentration in the muscles of female rats swimming in warm water was significantly lower (by approximately 16%, *p* ≤ 0.001) compared to those training in cold water (Figure 1). There were no statistically significant differences in the ADP concentration between female and male rats in groups of animals swimming in both warm and cold water.

### 2.3. AMP Concentration in the Muscle of Rats 

AMP concentration in the muscles of male rats undergoing training in cold water was significantly lower (by approximately 50%, *p* ≤ 0.001) compared to the content of this compound in the muscles of male rats from the control group. Similarly, in the muscles of male rats training in warm water, a significantly lower AMP concentration was observed (by approximately 42%, *p* ≤ 0.001) compared to the level found in the muscles of animals from the control group. The AMP concentration in the muscles of male rats swimming in warm water was also significantly higher than in the muscles of those training in cold water (by approximately 19%, *p* = 0.0018) (Figure 1).

In female rats undergoing training in cold water, the AMP concentration in their muscles was also significantly lower (by approximately 48%, *p* ≤ 0.001) compared to the control group. Similarly, female rats training in warm water exhibited a lower AMP concentration in their muscles (by approximately 34%, *p* ≤ 0.001) compared to the female control group. Furthermore, the AMP concentration in the muscles of female rats swimming in warm water was significantly higher (by approximately 28%, *p* ≤ 0.001) compared to those training in cold water (Figure 1). There were no statistically significant differences in the AMP concentration between female and male rats in groups of animals swimming in both warm and cold water.

### 2.4. Ado Concentration in the Muscle of Rats

Regarding Ado concentration in the muscles of male rats, those swimming in the 5 °C group had a significantly higher concentration (by approximately 48%, *p* ≤ 0.001) compared to the muscles of male rats from the control group. In the muscles of male rats swimming in warm water, a higher Ado concentration was also observed (by approximately 25%, *p* = 0.002) compared to the control group. Additionally, in the muscles of male rats swimming in warm water, a significantly lower Ado concentration was found compared to those training in cold water (by approximately 15%, *p* = 0.005) (Figure 1).

Similarly, in female rats from the 5 °C group, a higher Ado concentration in their muscles was found (by approximately 55%, *p* ≤ 0.001) compared to the control group. Female rats training in warm water also showed a higher Ado concentration in their muscles (by approximately 10%, *p* ≤ 0.003) compared to the female control group. Moreover, the Ado concentration in the muscles of female rats swimming in warm water was significantly lower (by approximately 29%, *p* ≤ 0.001) compared to those training in cold water (Figure 1). There were no statistically significant differences in the Ado concentration between female and male rats in groups of animals swimming in both warm and cold water.

### 2.5. TAN Concentration in the Muscle of Rats 

Concerning TAN concentration in the muscles of male rats, those swimming in cold water had a significantly higher concentration (by approximately 35%, *p* ≤ 0.001) compared to the muscles of male rats from the control group. In the muscles of male rats swimming in warm water, higher TAN concentrations were also observed (by approximately 4.6%, *p* = 0.009) compared to the control group. Additionally, in the muscles of male rats swimming in warm water, a significantly lower TAN concentration was found compared to those training in cold water (by approximately 23%, *p* ≤ 0.001) (Figure 2).

Similarly, in female rats training in cold water, a higher TAN concentration in their muscles was observed (by approximately 28%, *p* ≤ 0.001) compared to the control group. Female rats training in warm water also showed a higher TAN concentration in their muscles (by approximately 13%, *p* ≤ 0.001) compared to the female control group. Furthermore, the TAN concentration in the muscles of female rats swimming in warm water was significantly lower (by approximately 12%, *p* ≤ 0.001) compared to those training in cold water (Figure 2). There were no statistically significant differences in the TAN concentration between female and male rats in groups of animals swimming in both warm and cold water.

### 2.6. AEC Value in Rat Muscles

The AEC value in the muscles of male rats from the 5 °C group was significantly higher (by approximately 14%, *p* ≤ 0.001) compared to the concentration in the muscles of male rats from the control group. Similarly, in the muscles of male rats swimming in warm water, a higher AEC value was also observed (by approximately 7%, *p* ≤ 0.001) compared to the control group. The AEC value was significantly lower in the muscles of male rats training in warm water compared to the muscles of male rats training in cold water (by approximately 5.5%, *p* ≤ 0.001) (Figure 2).

In the muscles of female rats undergoing training in cold water, a higher AEC value was found (by approximately 6%, *p* ≤ 0.001) compared to the control group. Female rats training under thermoneutral conditions also showed a higher AEC value in their muscles (by approximately 4%, *p* ≤ 0.001) compared to the female control group. Furthermore, the AEC value in the muscles of female rats swimming in warm water was statistically significantly lower (by approximately 2%, *p* ≤ 0.001) compared to those training in cold water (Figure 2). There were no statistically significant differences in the AEC value between female and male rats in groups of animals swimming in both warm and cold water.

### 2.7. Expression of PGC-1α in Rat Muscles

The expression of the PGC-1α protein and mRNA in the muscles of male rats swimming in cold water was significantly higher (by approximately 34%, *p* ≤ 0.001 and 20%, *p* ≤ 0.001, respectively) compared to the expression in the muscles of male rats from the control group. This relationship was not observed in the group of male rats training in warm water compared to the control group. In male rats swimming in cold water, a significantly higher expression of the PGC-1α protein and gene was also found in their muscles compared to their expression in the muscles of male rats training in warm water (by approximately 16%, *p* ≤ 0.001 and 13%, *p* ≤ 0.001, respectively) (Figure 3).

Similarly, in female rats undergoing training in cold water, a higher expression of PGC-1α protein and mRNA was observed in their muscles (by approximately 23%, *p* ≤ 0.001 and 20%, *p* ≤ 0.001, respectively) compared to the control group. The expression of PGC-1α protein and mRNA in the muscles of female rats swimming in cold water was also significantly higher (by approximately 10%, *p* ≤ 0.001 and 14%, *p* ≤ 0.001, respectively) compared to those training in warm water. However, this relationship was not observed in the group of female rats training in warm water compared to the control group (Figure 3).

### 2.8. Expression of Mfn1 in Rat Muscles

The mRNA and protein expression of Mfn1 in the muscles of male rats swimming in cold water was significantly higher (mRNA 40%, *p* ≤ 0.007 and protein 55%, *p* ≤ 0.002, respectively) compared to its expression in the muscles of male rats from the control group. Furthermore, male rats training in cold water exhibited a significantly higher expression of the Mfn1 gene and protein in their muscles compared to those training in warm water (15%, *p* ≤ 0.002 and 13%, *p* ≤ 0.002, respectively) (Figure 4).

In female rats undergoing training in cold water, a higher mRNA expression of Mfn1 was found (75%, *p* ≤ 0.0002 and 38%, *p* ≤ 0.002, respectively) compared to the control group. Moreover, the mRNA expression of Mfn1 in the muscles of female rats swimming in cold water was also significantly higher (66%, *p* ≤ 0.001 and 28%, *p* ≤ 0.002) compared to those training in warm water (Figure 4).

### 2.9. Expression of Mfn2 in Rat Muscles

The mRNA and protein expression of Mfn2 in the muscles of male rats swimming in cold water was significantly higher (mRNA 29%, *p* ≤ 0.02 and protein 37%, *p* ≤ 0.002, respectively) compared to its expression in the muscles of male rats from the control group. This relationship was not observed in the group of male rats training in warm water compared to the control group. Furthermore, the protein expression of Mfn2 in the muscles of male rats swimming in cold water was also significantly higher (13%, *p* ≤ 0.001) compared to those training in warm water (Figure 5).

In female rats undergoing training in cold water, a higher mRNA and protein expression of Mfn2 was found (93%, *p* ≤ 0.002 and 22%, *p* = 0.004, respectively) compared to the control group. Furthermore, the mRNA and protein expression of Mfn2 in the muscles of female rats swimming in cold water was also significantly higher (38%, *p* ≤ 0.008 and 13% *p* = 0.004, respectively) compared to those training in warm water (Figure 5). 

The immunoexpression of mitofusins (Mfn1 and Mfn2) in the skeletal muscle of male and female rats from the control and experimental groups is shown in Figure 6 and Figure 7. The immunoexpression of Mfn1 among the skeletal muscles of male rats was the highest in the rats swimming in cold water, and it was also higher in the rats training in warm water, in comparison to the control group (Figure 6, upper panel). A similar result was observed in female rats; this means that the highest Mfn1 immunoexpression was in rats training in cold water, and this was found to be slightly higher in rats swimming in cold water in comparison to the control female rats (Figure 6, lower panel). 

The immunoexpression of Mfn2 in the skeletal muscles of male rats swimming in cold water was slightly higher than the control group and almost comparable to its expression in the muscles of male rats swimming in warm water (Figure 7, upper panel). In female rats undergoing training in cold as well as warm water, the immuoexpression of Mfn2 was relatively in the same level, but much higher than the group (Figure 7, lower panel). 

The level of Mfn1 immunoexpression was generally lower than Mfn2 and there was no detected sex-related expression of these mitofusins (Table 1). 

### 2.10. Expression of Opa1 in Rat Muscles

The mRNA and protein expression of Opa1 in the muscles of male rats swimming in cold water was significantly higher (mRNA 68%, *p* ≤ 0.02 and protein 37%, *p* = 0.019, respectively) compared to its expression in the muscles of male rats from the control group (Figure 8).

In female rats undergoing training in cold water, a higher mRNA and protein expression of Opa1 was found (mRNA 100%, *p* ≤ 0.001 and protein 61%, *p* = 0.024, respectively) compared to the control group. Additionally, the expression of Opa1 in the muscles of female rats swimming in cold water was also significantly higher (mRNA 90%, *p* ≤ 0.001 and protein 33%, *p* = 0.017, respectively) compared to those training in warm water (Figure 8). 

### 2.11. Expression of Drp1 in Rat Muscles

The mRNA and protein expression of Drp1 in the muscles of male rats swimming in cold water was significantly higher (mRNA 103%, *p* ≤ 0.02 and protein 52%, *p* ≤ 0.001, respectively) compared to its expression in the muscles of male rats from the control group. Additionally, the expression of Drp1 in the muscles of rats training in cold water was also significantly higher (77%, *p* = 0.048 and 19%, *p* = 0.022, respectively) than in the muscles of male rats training in warm water. In female rats undergoing training in cold water, a higher expression of the Drp1 gene and protein was found in the muscles (49%, *p* ≤ 0.001 and 37%, *p* = 0.03, respectively) compared to the control group. Similarly, in the muscles of female rats swimming in cold water and undergoing training, the expression of the Drp1 gene and protein was significantly higher (35%, *p* = *p* ≤ 0.001 and 15%, *p* = 0.007, respectively) compared to those training in warm water (Figure 9). 

## 3. Discussion

The conducted research on the impact of daily swimming in cold water (at a temperature of 5 °C) and in water with a comfortable thermal temperature (36 °C) over a period of 8 weeks on the parameters of the energy state, purine compound content, biogenesis, and the dynamics of mitochondria in the skeletal muscles of aging rats, showed a significant influence on the physical activity in the aquatic environment, concerning the energy status and factors regulating mitochondrial biogenesis in muscle tissue. This effect was expressed by higher concentrations of ATP, ADP, and Ado, a lower concentration of AMP, and increased AEC and TAN values, along with increased mRNA and protein expression of PGC-1α, Mfn1, Mfn2, Opa1, and Drp1. It is worth noting that, with respect to our hypotheses, this effect was significantly stronger in the case of activity in cold water compared to activity in water at a comfortable thermal temperature.

The direction of the observed impact of repeated swimming sessions in cold water on indicators of the energy state, purine compound concentration, and mitochondrial biogenesis was independent of the animals’ sex, resulting in an increase in all studied variables (except for a decrease in AMP). However, in the case of swimming training in water at a comfortable thermal temperature, the response was sex dependent. In males, swimming training resulted in an increase in ATP and Ado, a decrease in AMP, and no effect on ADP, along with an increase in AEC and TAN. In females, it led to an increase in ATP, ADP, and Ado, a decrease in AMP, and an increase in AEC and TAN. In the muscles of both males and females swimming in warm water, no changes in the mRNA and proteins associated with mitochondrial biogenesis and dynamics were observed. Therefore, it can be concluded that the additive effect of daily activity in cold water influenced both an increase in the rate of energy metabolism in the muscles of the studied animals and an intensification of mitochondrial biogenesis and dynamics (related to fusion and fragmentation processes). Daily activity in warm water also resulted in an increase in the rate of energy metabolism in muscles, but at the same time did not cause significant changes in mitochondrial dynamics.

### 3.1. Swimming in Cold Water

The existing literature provides no data on the impact of swimming training in cold water on energy metabolism, mitochondrial biogenesis, and dynamics in aging skeletal muscles, making it difficult to compare our results with those of other authors. However, swimming in cold water, also known as winter swimming or swimming in icy water, referring to swimming in the open air (in lakes, rivers, seas, or outdoor pools) mainly during winter in colder and polar regions, has been known for a long time [17]. In some northern countries such as Finland, Russia, Norway, Sweden, Denmark, Estonia, Lithuania, the Czech Republic, Latvia, and Poland, swimming in cold water is popular and regularly practiced during colder seasons [18]. In recent years, swimming in icy water (at temperatures below 5 °C) has become a year-round sport in which many swimmers regularly compete in local and international competitions [18,19,20]. Numerous studies have shown that swimming in cold water can bring many health benefits [21], including lowering blood pressure [22,23,24,25,26], hormonal balance [27,28,29], increased tissue sensitivity to insulin [21,27,30], reduced frequency and a milder course of upper respiratory tract infections [31,32,33], favorable effects on lipid metabolism [23,27,34,35], improvement of hematological parameters [36,37], stimulation of the immune system [38,39,40], alleviation of mood disorders [41], and the improvement of general well-being [42]. Numerous studies have highlighted the beneficial effects of swimming in cold water on the cardiovascular system and heart disease risk factors, including lipid profile, blood pressure, and various hormone levels and improving stress adaptation and insulin concentration [34,35,42,43]. Furthermore, exposure to low water temperatures (0–2 °C) for brief periods, three times a week for three winter months, resulted in reduced levels of adrenocorticotropic hormone (ACTH) and cortisol in the blood [44], and increased catecholamine concentration, which is potentially beneficial for treating depression, as it activates the sympathetic nervous system and raises norepinephrine and β-endorphin levels [41,45]. Previous studies have shown that individuals engaging in swimming in cold water experience a reduction in the frequency of upper respiratory tract infections [46,47]. Moreover, swimming in cold water has been found to have a positive impact on blood immunological parameters [38,48]. This improvement in immune response may be attributed to the release of stress hormones in response to cold exposure [27,49]. However, studies on the impact of swimming in cold water on leukocyte count and immunoglobulin concentration have yielded conflicting results. The likely cause of the discrepancies in the findings could be attributed to different research protocols. Some participants only immersed in icy water [20,50], some engaged in static swimming (remaining motionless in cold water for an extended period) [51,52], and some were long-distance swimmers, whose training (dynamic swimming in cold water) can last up to 6–8 h [31]. 

The resea0rch model used in this study was previously applied by Lubkowska et al. [53]. In this study male rats, physical effort in cold water resulted in a decrease in body mass compared to the pre-study values, which, as the authors suggest, was associated with increased energy expenditure on thermoregulatory processes. In female rats swimming in cold water, a decrease in body mass was observed only during the first two weeks of the experiment, followed by an increase until the end of the training, with the final body mass comparable to the initial values. In the case of swimming training at the thermal comfort temperature, a decrease in body mass was observed in both male and female rats, which was solely due to physical effort. As these presented studies are a continuation of the research conducted by the aforementioned authors and were performed on available tissues from rats from the cited experiment, they can provide valuable data for analyzing the impact of swimming training in cold water on the energetic metabolism of muscles, particularly in the context of aging. Short-term exposure to low temperatures is used as a treatment to counteract aging. However, the effect of cold on the body as a preventive measure requires more detailed research, especially in older individuals [54]. The results of Dudzinska and Lubkowska’s [55] research on the impact of a single exposure to cold applied systemically at −130 °C confirmed that this treatment leads to changes related to energy metabolism within red blood cells. These changes are beneficial from a cellular bioenergetics perspective, as they result in an increase in intracellular ATP and AEC concentrations in the red blood cells. Additionally, the metabolic response of purines to cryogenic temperatures is also associated with a decrease in the concentration of purine catabolism products, namely inosine and hypoxanthine in the blood. This suggests that the increased ATP concentration in red blood cells, along with the reduced levels of inosine and hypoxanthine in the plasma, provide a strong basis for hypothesizing an increase in energy resources in other cells as well.

In our study, daily swimming activity in cold water repeated for 8 weeks led to an increase in ATP production, improved parameters of the energy state, and beneficially affected mitochondrial dynamics in the skeletal muscles of aging rats. Therefore, it can be assumed that repeated exposure to cold may be a factor leading to adaptive changes that affect the overall state of the body and slow down aging. The research results of Hoffman-Goetz et al. [56] suggest that physical effort in conditions of cold exposure may increase cold tolerance in aging mice, but it is unclear whether this effect could delay aging. However, it has been found that physical effort increases tissue sensitivity to insulin and supports thermogenesis in brown adipose tissue in aging rats [56]. Hence, it is plausible to assume that increased insulin resistance may disrupt thermogenesis, while physical effort, by reducing the body’s resistance to insulin, may improve its cold tolerance [57,58]. Arnold et al. [59] hypothesized that the increased production of energy in the form of heat through an accelerated metabolism in rats subjected to physical effort in a cold environment is sufficient to compensate for the impact of the stressful factor that is cold, without triggering non-shivering thermogenesis in brown adipose tissue. The observed increase in ATP concentration, improvement in energy parameters (AEC, TAN values), and enhanced mitochondrial biogenesis in the skeletal muscles of aging rats subjected to swimming training in cold water, as presented in this study, can support this view.

### 3.2. Mitochondrial Biogenesis and Energy Metabolism

Biogenesis of mitochondria, a highly coordinated process involving transcriptional gene activation, lipid and protein synthesis, and the coordinated assembly of protein complexes into a functional respiratory chain, is a key element in improving muscle performance in response to exercise [60,61,62,63]. These complex processes result in an increase in the total amount of mitochondria within muscles and an expansion of the mitochondrial network, which improves the cell’s ability to adjust ATP production and hydrolysis, thus minimizing disruptions to cellular homeostasis that occur during the initial phase of muscular exertion.

Mitochondria from trained animals exhibited higher respiratory control and tightly coupled oxidative phosphorylation compared to mitochondria from untrained animals. The increased electron transport capacity was associated with enhanced ATP production, contributing to the prolonged endurance observed in trained animals after exercise. The key feature of the training protocol was intense interval exercise, based on the idea that each exercise cycle needed to exceed the minimum “threshold stimulus” to induce metabolic and morphological adaptations in muscles [63,64,65,66]. A crucial discovery was that the increased mitochondrial enzyme activity after training did not accompany a rise in cytosolic creatine kinase and adenylate kinase activity, validating the notion that improved ATP synthesis through oxygen-dependent pathways was the defining feature of physically trained muscles [67]. Subsequent human studies confirmed this hypothesis. Long-term physical training [68] and cross-sectional comparisons between trained and untrained individuals revealed higher maximal mitochondrial enzyme activity in quadriceps muscle samples [69] and a greater number of mitochondria visualized through electron microscopy techniques [70]. Further investigations demonstrated that endurance training increased mitochondrial content, mitochondrial enzyme activity, and VO_2max_, but these parameters were lower in older individuals [71]. The effect was attenuated in older individuals, indicating age-related differences in the response to training [71].

For the first time, observations suggested that the “plasticity” of muscles may decrease with aging. These discoveries contributed to the understanding that aerobic capacity is not only limited by factors that regulate oxygen supply (i.e., cardiovascular and respiratory factors). However, these findings also led to the belief that the aerobic capacity of the whole body was limited not only by the oxygen delivery/transport system (i.e., “central” cardiovascular and respiratory factors) but also by the number of skeletal muscle mitochondria, according to the function of the working muscle as the final recipient of oxygen during ATP production. Studies on rodents allowed the development of a model showing the relationship between the number of mitochondria and metabolic control in skeletal muscles during physical exertion. Studies conducted mainly on rodents contributed to the creation of a model proposed that a greater number of mitochondria improves oxidative sensitivity to ADP, and the rate of aerobic ATP production required to support muscle contraction would require a smaller increase in ADP at the beginning of exertion due to a larger number of mitochondria [72,73,74].

One of the early studies aiming to indicate the molecular mechanisms of increased mitochondrial protein content due to muscle contractions was conducted by Williams et al. [75]. They observed a higher expression of cytochrome b mRNA, increased citrate synthase activity, and mitochondrial DNA content (as markers of mitochondrial quantity) after chronic electrical stimulation of the rabbit tibialis anterior muscle. Subsequent studies revealed that various families of mitochondrial proteins are regulated by transcription factors, among which PGC-1α plays a key role in coordinating the expression of mitochondrial proteins and cooperating with PPARγ receptor in the regulation of the expression of many different genes and mitochondrial genomes [76].

PGC-1α is known to be induced mainly during increased oxygen demand in tissues, including increased physical exercise in skeletal muscles [16], and exposure to cold in brown adipose tissue [77]. The presented research also observed a significant increase in PGC-1α mRNA and protein expression in the muscles of rats swimming in cold water compared to the control group. In the group of rats swimming in thermoneutral conditions, a tendency for increased mRNA and protein expression of PGC-1α was also observed, but this change was not statistically significant. Several studies have shown that a single session of endurance exercise leads to an increase in mRNA and protein expression of PGC-1α both in rodents [78] and in human skeletal muscles [79]. Intensive training also increased mRNA and the protein expression of PGC-1α in skeletal muscles of rodents and humans [80,81,82,83]. It has been suggested that PGC-1α is a key regulator of mitochondrial biogenesis in response to neuromuscular stimuli and prevailing contractile activity.

Additionally, the tumor suppressor protein p53 has been identified as another transcription factor playing a role in exercise-induced mitochondrial biogenesis in skeletal muscles. Studies have shown that p53 knockout mice exhibited decreased endurance compared to wild-type mice, along with reduced mitochondrial content, decreased PGC-1α expression, and a lack of mitochondrial biogenesis-related transcripts after exercise [83]. Therefore, p53 is likely activated by AMPK and/or p38 MAPK and may act as an inducer of enzymes involved in mitochondrial electron transport. AMPK, a member of protein kinases detecting metabolites, acts as a metabolic “fuel gauge” in skeletal muscles, consisting of regulatory α, β, and γ subunits that exist in multiple isoforms and are required for full enzymatic activity. AMPK has been shown to be activated in response to exercise-induced elevation of AMP levels [84] and its activation has been found to depend on exercise intensity [85]. Rapid and transient AMPK activation after intense exercise has been associated with a rapid increase in AMP concentration in muscles. After a period of training, lower AMP concentrations and weaker AMPK activation were observed, possibly explaining the weakened activation during exercise performed at the same absolute (pre-training) muscle work rate [86,87].

In the presented research, a decrease in AMP concentration was observed in the muscles of aging rats subjected to training in cold water and swimming in thermoneutral water, along with increased mRNA and protein expression of PGC-1α in the muscles of rats training in cold water. The lowest AMP concentration was observed during swimming in cold water, indicating a state of mitochondrial biogenesis equilibrium resulting from the adaptation process. The increased number of mitochondria may contribute to the increased ATP concentration and energy charge value in the muscles, especially during training in cold water.

The observed significant increase in mRNA and the protein expression of Mfn1, Mfn2, Opa1, and Drp1 may indicate intensified fusion and fission processes, and thus increased mitochondrial dynamics in muscles. Increased energy demand may influence the formation of mitochondrial networks, which may occur when cells are exposed to cold, optimizing their functions under these stressful conditions. Increased mitochondrial dynamics may represent an adaptation to changing environmental conditions, positively influencing the metabolic state of muscles and contributing to increased ATP synthesis, as observed in the presented research.

## 4. Materials and Methods

### 4.1. Animals

Procedures involving animals were conducted in strict adherence to international animal care guidelines, with a focus on minimizing suffering and the number of animals used. The experiments received approval from the Local Ethical Committee on Animal Testing at the Pomeranian Medical University in Szczecin, Poland (approval No. 4/2014).

The study involved 64 Wistar rats of both genders (32 males and 32 females), aged 15 months at the start of the experiment. Prior to the research, the animals underwent a one-month quarantine period in the animal facility. During both the quarantine and experimental phases, the rats were provided with standard laboratory chow (Murigran pellet, Motycz, Poland) as their diet and had unrestricted access to tap water for drinking. The rats were randomly divided into the following groups:Control groups (n = 16 animals)-animals kept in sedentary conditions: Control group males (n = 8); Control group females (n = 8)Study groups 5 °C (n = 24 animals)-animals underwent swimming training in cold water at 5 ± 2 °C: Group 5 °C males (n = 12); Group 5 °C females (n = 12)Study groups 36 °C (n = 24 animals)-animals underwent swimming training in water with a thermal comfort temperature of 36 ± 2 °C: Group 36 °C males (n = 12); Group 36 °C females (n = 12)

Throughout the study, all the rats were housed in specialized plastic cages with open work covers and solidified wood dust as litter. They were bred in pairs in conventional polipropylen cages (ANIMALAB, Poznań, Poland). The animal facility maintained a constant temperature (23 ± 2 °C), air humidity (approximately 40%), and a 12 h day and night cycle. Four animals from the same group shared a single cage with free access to dry food and water.

### 4.2. Experimental Procedure

The animals in the experiment underwent a series of swimming sessions over 9 weeks. Before the swimming sessions began, each rat from the control and study groups was introduced to an empty glass tank (dimensions: length 100 cm, width 50 cm, depth 50 cm) for 2 min to familiarize them with the training sessions and the experimental environment. This process was repeated for 7 days before the swimming sessions commenced.

During the first week, the duration of the initial swim was 2 min and gradually increased by 0.5 min per day, reaching 4 min on the fifth day. This duration was based on the findings of previous researchers, who demonstrated that rats actively swam for up to 4 min in water at 4–5 °C and then began to drift [48]. Considering literature data indicating that rats could remain active for longer in thermally comfortable conditions [37], a 4 min daily swimming session was chosen for both study groups. This way, the water temperature in the glass tank was the only differing variable between the study groups.

From the second to the eighth week of the experiment, each swimming session lasted 4 min per day. The sessions were conducted every day, five days a week, in the morning hours, concluding by 12 a.m. Only one rat swam in each glass tank at a time. After each swimming session, the rats were carefully dried with a paper towel, wrapped in dry lignin in separate cages for a few minutes to prevent excessive chilling, and then placed back in their home cages. Rats in the control group were also familiarized with the experimenter daily and placed in an empty glass tank for 4 min.

At the end of the study, 48 h after the last swimming training session, the rats were euthanized under anesthesia using ketamine hydrochloride/xylazine (100/10 mg/kg body weight administered intraperitoneally). Rats’ medial gastrocnemius muscle was collected during the animal dissection and immediately stored in liquid nitrogen. The samples were subsequently kept at −80 °C until further analysis of ATP, ADP, AMP, Ado concentrations, and the mRNA and protein expression of PGC-1α, Mfn1, Mfn2, Opa1, and Drp1.

### 4.3. Determination of ATP, ADP, AMP, and Ado Concentrations Using High-Performance Liquid Chromatography (HPLC)

The concentration of ATP, ADP, AMP, and Ado in the muscles was analyzed using the high-performance liquid chromatography (HPLC) method according to Smolenski et al. [88]. Frozen muscle tissue in liquid nitrogen was placed in a thermobox (−21 °C). Then, a small tissue fragment was transferred to a metal homogenizer (previously cooled in a container with liquid nitrogen) and was covered with 2–3 portions of liquid nitrogen. The homogenization was achieved through repeatedly striking the tissue with a hammer (4–5 times) on a metal pin (also previously cooled in a container with liquid nitrogen). The powdered and frozen muscle tissue (approximately 1 mg of protein) was transferred to an Eppendorf tube containing 500 µL of 1.3 M perchloric acid (previously cooled to 4 °C). After brief mixing (vortex), further homogenization was carried out using a knife homogenizer for about 15 s. The homogenate was then centrifuged (16,000× *g* for 10 min at 4 °C). The resulting supernatant (600 µL) was neutralized by adding approximately 60–90 µL of a 3 M solution of potassium orthophosphate (pH 6.0–7.0). The neutralized supernatant was centrifuged under the same conditions as before. The obtained clear filtrate was stored at −80 °C until further analysis. The prepared samples were used to determine the concentration of purine derivatives. The analytical procedure was carried out using the Hewlett Packard Series 1100 chromatographic system.

### 4.4. Analysis of PGC-1α, Mfn1, Mfn2, Opa1, and Drp1 Gene Expression Using Real-Time Polymerase Chain Reaction (qRT-PCR)

Quantitative evaluation of the expression of PGC-1α (Ppargc1a), Mfn1, Mfn2, Opa1, and Drp1 genes encoding peroxisome proliferator-activated receptor gamma coactivator (PGC-1α), mitofusin 1 (Mfn1), mitofusin 2 (Mfn2), optic atrophy 1 (Opa1), and dynamin-related protein 1 (Drp1) in the muscles was conducted using quantitative real-time polymerase chain reaction (qRT-PCR). RNA was isolated from tissues stored at −80 °C using a commercial RNeasy Mini Kit (Qiagen, Hilden, Germany). The quantity and quality of the isolated RNA were assessed using a NanoDrop ND-1000 spectrophotometer (NanoDrop Technologies, Wilmington, DE, USA). The RNA was transcribed into cDNA using 1 µg of RNA in a total sample volume of 20 µL with Omniscript RT Kit (Qiagen, Germany) following the manufacturer’s instructions. qRT-PCR was conducted using the 7500 Fast Real-Time PCR System (Applied Biosystems, Foster City, CA, USA) and Power SYBR Green PCR Master Mix (Applied Biosystems, USA), which is a buffer solution containing AmpliTaq Gold DNA polymerase, a mixture of deoxynucleotides, SYBR Green dye, and the reference dye ROX (for fluorescence signal normalization). Monitoring of the PCR reaction product amplification in real-time was achieved by measuring fluorescence, which is proportional to the concentration of the product in the mixture. Each sample was analyzed twice, and the results were averaged from two measurements. The reaction mixture consisted of 12.5 µL of Power SYBR Green PCR Master Mix, forward and reverse primers for the target gene at a final concentration of 0.5 µM, and 50 ng of cDNA. The total reaction volume was 25 µL. The reaction conditions were as follows: 95 °C (15 s), 40 cycles at 95 °C (15 s), and 60 °C (1 min). The expression level of the genes in each sample was compared with the endogenous control. The specificity of the reaction was assessed using melting curve analysis, confirming the amplification of only one PCR reaction product. The following primers were used in the study:name: Gapdh_Fsequence: ATGACTCTACCCACGGCAAGname: Gapdh_Rsequence: CTGGAAGATGGTGATGGGTTname: Ppargc1a_Fsequence: TATGGAGTGACATAGAGTGTGCTname: Ppargc1a_Rsequence: GTCACTACACCACTTCAATCCname: Mfn1_Fsequence: ATGGCAGAAACGGTATCTCCAname: Mfn1_Rsequence: GCCCTCAGTAACAAACTCCAGTname: Mfn2_Fsequence: AGAACTGGACCCAGTTACTAname: Mfn2_Rsequence: CACCTCGCTGATTCCCCTGAname: Opa1_Fsequence: CCGTGTGAGCAGAAGAACACname: Opa1_Rsequence: AGCCTCAAGGCCAACTATGTname: Drp1_Fsequence: CAGGAACTGTTACGGTTCCCTAAname: Drp1_Rsequence: CCTGAATTAACTTGTCTCGCGA

The relative expression of all the genes was determined using the comparative ΔΔCt method, according to the formula R = 2^−ΔΔCt^, where ΔCt for the calibrator = (Ct of the calibrator for the target gene) − (Ct of the reference gene), ΔCt for the unknown sample = (Ct of the target gene) − (Ct of the reference gene), and ΔΔCt = (ΔCt of the sample for the target gene) − (ΔCt of the calibrator for the target gene), where Ct represents the cycle threshold.

### 4.5. Analysis of PGC-1α, Mfn1, Mfn2, Opa1, and Drp1 Protein Expression Using ELISA Method

In order to analyze protein expression, the muscles tissue was homogenized using knife homogenization of approximately 1 cm^3^ of tissue in 1 mL of commercial RIPA (Thermo Fisher Scientific, Waltham, MA, USA) containing proteinase inhibitors (PhosSTOP and cOmplete, Mini Protease Inhibitor Cocktail, Sigma-Aldrich, Poznań, Poland). The entire process was carried out at a temperature of 4 °C. In the next step, the samples were centrifuged (5 min at 5000× *g*) to obtain the supernatant. In a further step, the concentration of total protein in the supernatant was determined using MicroBCAPierce™ (Thermo Fisher Scientific, Waltham, MA, USA).

PGC-1α, Mfn1, Mfn2, Opa1, and Drp1 protein expression determinations were performed using commercial ELISA test kits: Rat PGC1-α (Peroxisome proliferator-activated receptor gamma coactivator 1-alpha) ELISA Kit (MBS288117, MyBioSource, Inc., San Diego, CA, USA), (Range 0.156–10 ng/mL, Sensitivity < 0.078 ng/mL); Rat Dnm1l (Dynamin-1-like protein) ELISA Kit (ER6426, FineTest, Wuhan, China), (Range 78.125–5000 pg/mL, Sensitivity < 46.875 pg/mL); Rat OPA1 Sandwich ELISA Kit (LS-F74404, LSbio, Seattle, WA, USA), (Range 0.156–10 ng/mL, Sensitivity < 0.156 ng/mL); Rat Mitofusin 1 (MFN1) ELISA Kit abx 259144, Abbexa Ltd., Cambridge Science Park, Cambridge, UK), (Range0.156 ng/mL–10 ng/mL, Sensitivity < 0.06 ng/mL), Rat Mitofusin 2 (MFN2) ELISA Kit (abx259143, Abbexa Ltd., Cambridge Science Park, Cambridge, UK), (Range0.156 ng/mL–10 ng/mL, Sensitivity < 0.06 ng/mL). The results were read using a plate reader (BiochromAsys UVM 340, Biocompare, Poznań, Poland). The obtained values were converted into the amount of total protein.

### 4.6. Immunohistochemical (IHC) Analysis

The dissected muscle specimens were fixed in a 4% neutral buffered formalin solution for 24 h. Subsequently, they underwent a series of washes with distilled water, ethanol, and methanol to remove any residual fixative. The tissues were then dehydrated using a series of washes with absolute ethanol and xylene. After saturation with liquid paraffin, the samples were embedded in paraffin blocks. Serial sections 3–5 μm in thickness were prepared from the paraffin-embedded tissues using a microtome (MICROM HM340E, RWD, Poznań, Poland) and placed on silane-coated histological slides (3-aminopropyl-triethoxysilane, Thermo Scientific, Waltham, MA, USA). The sections were deparaffinized using xylene and ethanol in decreasing concentrations. To expose the epitopes, the deparaffinized sections were subjected to heat-induced antigen retrieval using a microwave oven in citrate buffer (pH 6.0). After cooling and washing with PBS, the sections were incubated with primary antibodies against the mitofusins (Mfn1 and Mfn2; Santa Cruz Biotechnology, Dallas, Texas, USA, cat. no.: sc-166644 and sc-515647, respectively, diluted 1:250) for 60 min at room temperature (RT). The antibodies were diluted in antibody diluent with background reducing components (cat. no.: K8024, Dako, Agilent Technologies, Santa Clara, CA, USA). To visualize the antigen–antibody complex, the Dako EnVision^®^ + Dual Link System-HRP (DAB+) (DakoCytomation, Agilent Technologies, Santa Clara, CA, USA, cat. no.: K4065) was used, following the included staining procedure instructions. The sections were washed in distilled H_2_O and counterstained with hematoxylin. Negative controls were processed without the primary antibody. Positive staining was determined via visual identification of brown pigmentation using a microscope (Leica DM5000B, Wetzlar, Germany).

### 4.7. Statistical Analysis

The obtained experimental results were subjected to statistical analysis using Excel and Statistica v10 programs. The Shapiro–Wilk test was conducted to assess whether the distributions of values for the analyzed variables deviated from a normal distribution. Mean values, standard deviations, medians, minimum and maximum values, and quartiles were calculated. The Mann–Whitney U test was applied to demonstrate the significance of differences. A significance level of *p* ≤ 0.05 was considered statistically significant.

## 5. Conclusions

Based on the results of this study, the following conclusions can be drawn: (i) physical exercise in cold water may have a positive impact on the energetic metabolism, biogenesis, and dynamics (fusion and fission processes) of aging rat muscle mitochondria; (ii) increased mitochondrial dynamics during physical exercise in cold water may improve mitochondrial function and optimize their bioenergetic capacities in aging rat muscles; (iii) changes in the concentration of high-energy compounds and the expression of protein regulating mitochondrial dynamics in muscles can serve as useful indicators for monitoring adaptive changes occurring in aging muscles under the influence of physical exercise in cold water.

### Limitation of the Study

Our study of energy metabolism in the muscles of aging rats exposed to cold-water swimming training has provided strong evidence supporting improvements in muscle energy metabolism, biogenesis, and mitochondrial dynamics. These improvements were observed at the mRNA as well as the protein expression level (using and/or Western blotting/immunolabeling/ELISA methods). Extending our investigation to protein expression using immunolabeling methods for all studied proteins would offer additional confirmation of heightened energy metabolism in aging rat muscle mitochondria. Additionally, a valuable direction for our upcoming research could involve exploring mitochondrial content, inner mitochondrial membrane potential, mitochondrial respiration, autophagy, mitophagy, and reactive oxygen species production. The results from these investigations, combined with the measured concentrations of high-energy compounds and the expression of proteins that regulate mitochondrial dynamics in muscles, could serve as practical indicators for monitoring adaptive changes in aging muscles in response to physical exertion in cold water conditions.

## Figures and Tables

**Figure 1 ijms-25-04055-f001:**
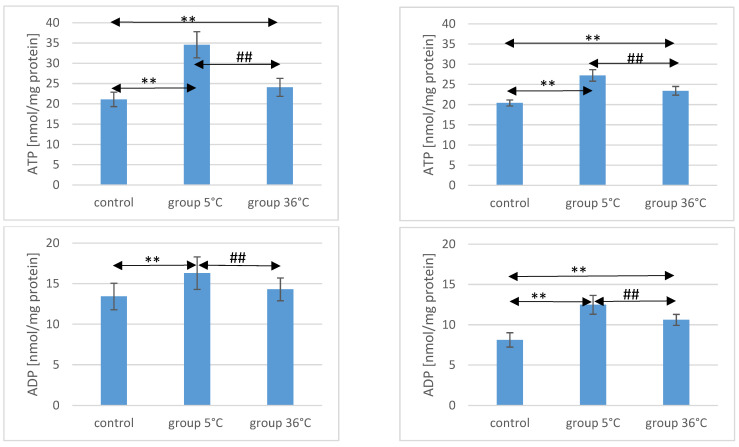
ATP, ADP, AMP, and Ado in the muscles of male (**A**) and female (**B**) rats from the control group and the experimental groups. The results are presented as means and standard deviations. ** *p* < 0.005 level of significance compared to the control group (Mann–Whitney U test), ## *p* < 0.005 level of significance compared to the 5 °C group (Mann–Whitney U test).

**Figure 2 ijms-25-04055-f002:**
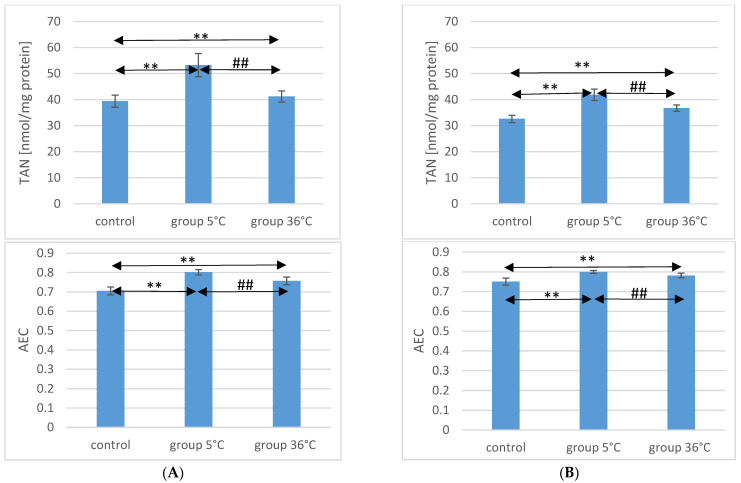
TAN and AEC in the muscles of male rats (**A**) and female rats (**B**) from the control group and experimental groups. The results are presented as means and standard deviations. ** *p* < 0.005 level of significance compared to the control group (Mann–Whitney U test), ## *p* < 0.005 level of significance compared to the 5 °C group (Mann–Whitney U test).

**Figure 3 ijms-25-04055-f003:**
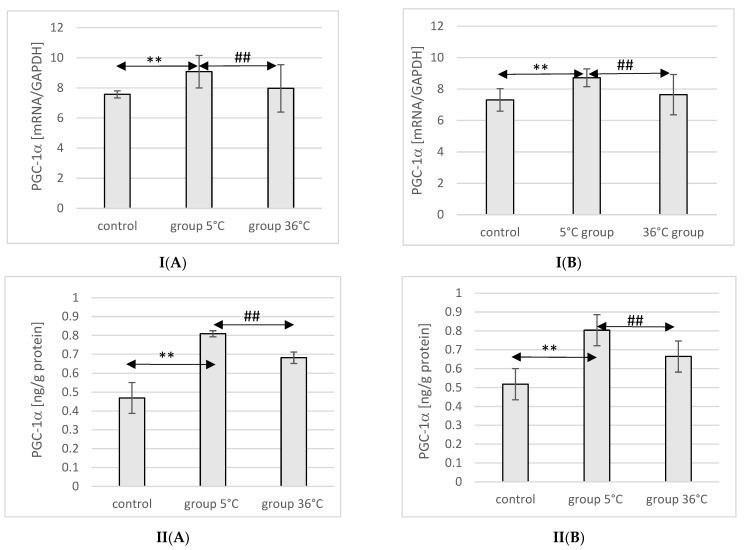
Expression of PGC-1α mRNA (**I**) and ELISA method protein analysis (**II**) in the muscles of male rats (**A**) and female rats (**B**) from the control group and experimental groups. The results are presented as means and standard deviations. ** *p* < 0.005 level of significance compared to the control group (Mann–Whitney U test), ## *p* < 0.005 level of significance compared to the 5 °C group (Mann–Whitney U test).

**Figure 4 ijms-25-04055-f004:**
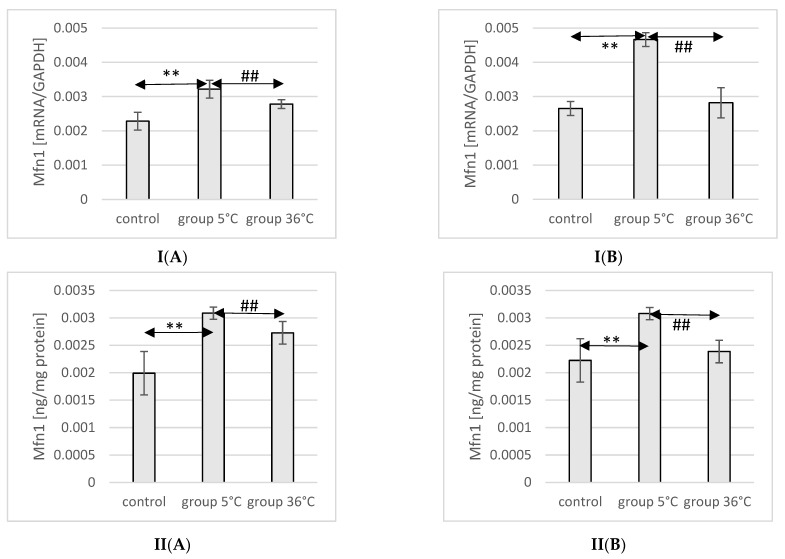
Expression of Mfn1 mRNA (**I**) and ELISA method protein analysis (**II**) in the muscles of male rats (**A**) and female rats (**B**) from the control group and experimental groups. The results are presented as means and standard deviations. ** *p* < 0.005 level of significance compared to the control group (Mann–Whitney U test), ## is *p* < 0.005 level of significance compared to the 5 °C group (Mann–Whitney U test).

**Figure 5 ijms-25-04055-f005:**
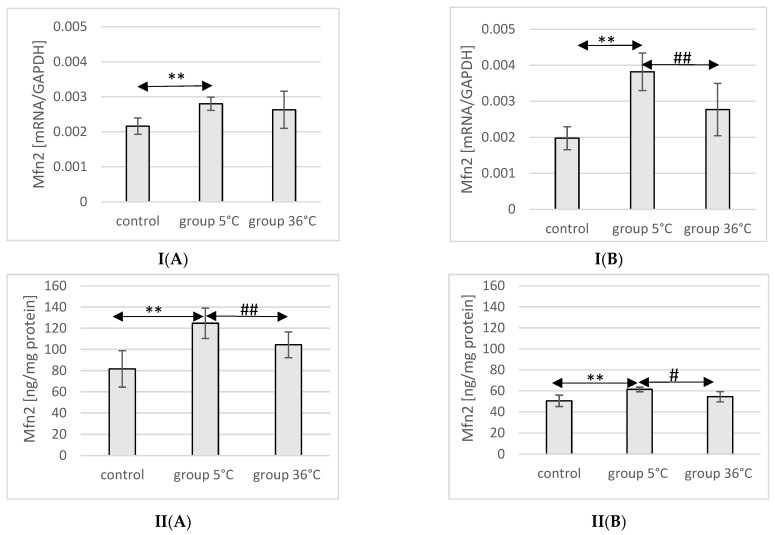
Expression of Mfn2 mRNA (**I**) and ELISA method protein analysis (**II**) in the muscles of male rats (**A**) and female rats (**B**) from the control group and experimental groups. The results are presented as means and standard deviations. ** *p* < 0.005 level of significance compared to the control group (Mann–Whitney U test), # is *p* < 0.05 level of significance compared to the 5 °C group (Mann–Whitney U test), ## is *p* < 0.005 level of significance compared to the 5 °C group (Mann–Whitney U test).

**Figure 6 ijms-25-04055-f006:**
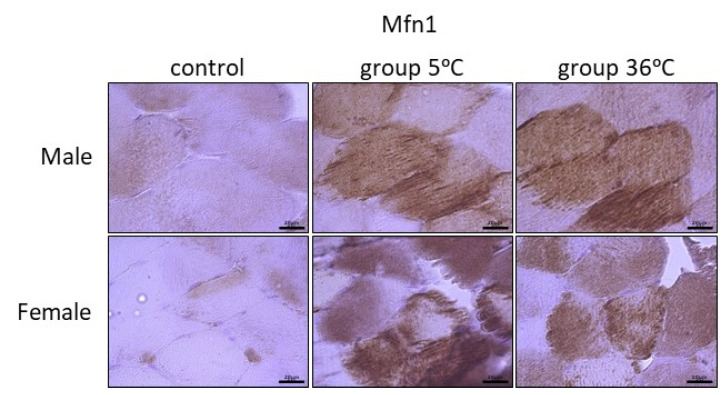
Representative microphotography showing immunoexpression of Mfn1 in muscles of male and female rats from the control group and experimental groups. The immunopositive reaction’s area appears as brown colored precipitates within muscle fiber. The color intensity of the precipitate indicates the level of immunoexpression of Mfn1 detected using IHC reaction. Scale bar 20 μm (objective magnification ×100).

**Figure 7 ijms-25-04055-f007:**
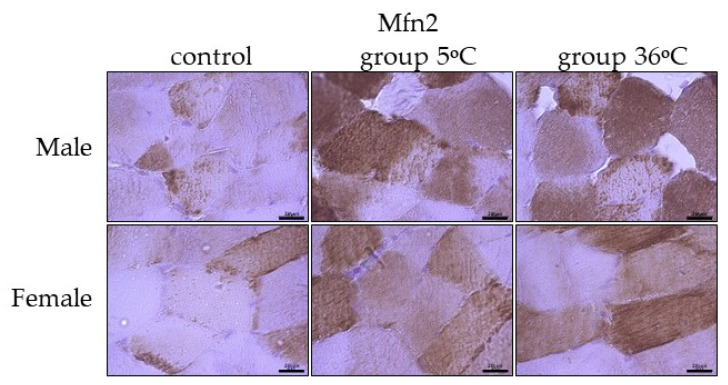
Representative microphotography showing immunoexpression of Mfn2 in the skeletal muscles of male and female rats from the control group and experimental groups. The immunopositive reaction’s area appears as brown colored precipitates within muscle fiber. The color intensity of the precipitate indicates the level of immunoexpression of Mfn2 detected using IHC reaction. Scale bar 20 μm (objective magnification ×100).

**Figure 8 ijms-25-04055-f008:**
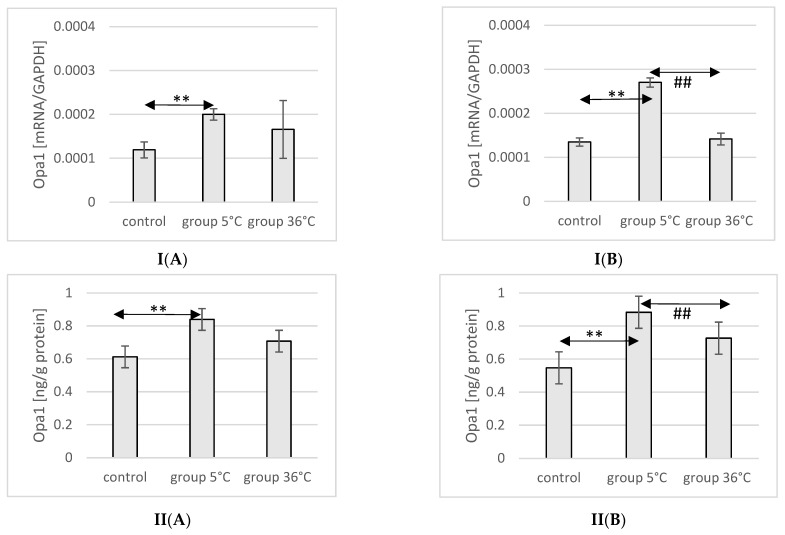
Expression of Opa1 mRNA (**I**) and ELISA method protein analysis (**II**) in the muscles of male rats (**A**) and female rats (**B**) from the control group and experimental groups. The results are presented as means and standard deviations. ** *p* < 0.005 level of significance compared to the control group (Mann–Whitney U test), ## is *p* < 0.005 level of significance compared to the 5 °C group (Mann–Whitney U test).

**Figure 9 ijms-25-04055-f009:**
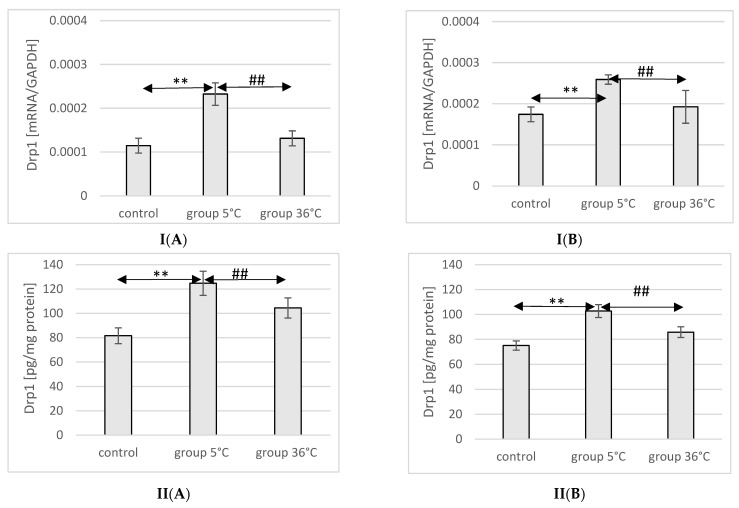
Expression of Drp1 mRNA (**I**) and ELISA method protein analysis (**II**) in the muscles of male rats (**A**) and female rats (**B**) from the control group and experimental groups. The results are presented as means and standard deviations; ** *p* < 0.005 level of significance compared to the control group (Mann–Whitney U test), ## is *p* < 0.005 level of significance compared to the 5 °C group (Mann–Whitney U test).

**Table 1 ijms-25-04055-t001:** Summary of the expression of Mfn1 and Mfn2 in the control and study groups presented as intensity of immunostaining.

	Male			Female		
	**Control**	**Group 5 °C**	**Group 36 °C**	**Control**	**Group 5 °C**	**Group 36 °C**
**Mfn1**	**+/−**	**+++**	**++**	**+/−**	**++**	**++**
**Mfn2**	**+**	**++**	**++**	**+**	**++**	**++**

Intensity of immunostaining scored as very weakly positive (+/-), weakly positive (+), moderately positive (++), strongly positive (+++).

## Data Availability

The data presented in this study are available on request from the corresponding author.

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
