# Peer review of "The Effect of Cold-Water Swimming on Energy Metabolism, Dynamics, and Mitochondrial Biogenesis in the Muscles of Aging Rats"

_ijms, 2024, doi:10.3390/ijms25074055_

Round 1

Reviewer 1 Report (New Reviewer)

Comments and Suggestions for Authors

The article is well-written in general, but I do not see much novelty for the article topic, as the low temperature swimming increases metabolism has been a topic that was discussed and well-established long time ago.  For example, an article published in 1987, Int J Sports Med. 1987 Oct;8(5):352-6, discussed the relationship between metabolic and hormonal response to swimming in cold water.

Besides, two minor issues for the paper to be fixed:

1) Figure 3. The beta-actin label for the Western blot results is off. 

2) The gene symbols should all be italicized, please check and correct all the gene symbols in the article. 

Comments on the Quality of English Language

The English of the paper is good overall. Special attention needs to be paid to the Nomenclatures for gene and protein names. 

Author Response

Review 1

Comments and Suggestions for Authors

The article is well-written in general, but I do not see much novelty for the article topic, as the low temperature swimming increases metabolism has been a topic that was discussed and well-established long time ago.  For example, an article published in 1987, Int J Sports Med. 1987 Oct;8(5):352-6, discussed the relationship between metabolic and hormonal response to swimming in cold water.

Thank you very much for this comment. We would like to point out that our manuscript does not only concern general hormonal changes and improvement of metabolism, but also points for the first time to the mechanism related to the mitochondria themselves, and more specifically to the processes of mitochondrial network formation and mitochondrial dynamics. Therefore, we believe that this is a very innovative approach to the impact of physical exercise in cold water on mitochondria.

Besides, two minor issues for the paper to be fixed:

1) Figure 3. The beta-actin label for the Western blot results is off. 

Thank you very much for this remark. In our study we used ELISA method and WB method.  Both techniques we used allow us to determine the amount of protein, however the ELISA method is a quantitative method and the WB method is a semi-quantitative one. For this reason, we decided not to present the results using the WB technique and presented only ELISA results for all tested proteins. We add determinations of PGC 1alpha protein level (previously examined by us using the WB technique).

 2) The gene symbols should all be italicized, please check and correct all the gene symbols in the article. 

All gene symbols were checked and corrected.

Comments on the Quality of English Language

The English of the paper is good overall. Special attention needs to be paid to the Nomenclatures for gene and protein names.

All gene symbols were checked and corrected.

Reviewer 2 Report (New Reviewer)

Comments and Suggestions for Authors

The authors conducted a rat study to examine the impact of cold-water swimming on the energy metabolism and mitochondrial dynamics and biogenesis in the aging rats. They found the increased gene and protein indicators of mitochondrial biogenesis and dynamics as well as the metabolites. The manuscript is well written and the study design in which they included sedentary control seems reasonable although they did not preclude the potential impact of the cold environment alone from the swimming exercise. I have a several minor comments on this manuscript.

1. What is the average lifespan for the Wistar rat? The authors used 15 months-old rats. Can the author call the 15 months old in rats aged when considering their lifespan and its translated age in the human?  

2. Abstract needs to be improved. Based on the way the authors represent their results, there are many variables which are significantly different between cold-water and thermal comfort groups. Also, compared to the sedentary control group, the magnitude of the difference is greater in the cold-water group than the thermal comfort group. However, the abstract does not reflect this aspect. Please revise the abstract appropriately.

3. Line 17, the authors may want to add detailed information about the age and the number of animals.

4. Line 23, It is confusing to recognize the TAN as the adenylate nucleotide pool. The authors may want to come up with clearer abbreviations.

5. Introduction is lengthy. Some of the background needs to go either methods or discussion section. One example, the detailed measurement outcome information (line 96-101) may not be necessary.

6. Figure legends, the author may want to modify # p level of significance compared to the 36°C group rather than 5°C group.

7. Figures 5 and 6 seem to show the cross-section of the muscle fiber, but I cannot recognize the clear section in most panels. Is this due to the cutting, processing, staining, or imaging problem? The authors need to replace the figures, which show clear cross-section.

8. Figure 8 looks like a hand drawn illustration, which is not appropriate for the publication. Please utilize the standardized tools for the overall scheme.

Author Response

Review 2

The authors conducted a rat study to examine the impact of cold-water swimming on the energy metabolism and mitochondrial dynamics and biogenesis in the aging rats. They found the increased gene and protein indicators of mitochondrial biogenesis and dynamics as well as the metabolites. The manuscript is well written and the study design in which they included sedentary control seems reasonable although they did not preclude the potential impact of the cold environment alone from the swimming exercise. I have a several minor comments on this manuscript.

  1. What is the average lifespan for the Wistar rat? The authors used 15 months-old rats. Can the author call the 15 months old in rats aged when considering their lifespan and its translated age in the human? 

Thank you very much for this remark. Our rats live for about 1.5 years (18 months, approx. 558 days), which means that at the age of 15 months (approx. 465 days) they are in the aging stage (midle adulthood 300-600 days), if translates age in the humans rats are approximately 40-65 years old. (Ghasemi A, et al. The laboratory rat: Age and body weight matter. EXCLI J. 2021. PMID: 34737685).   

  1. Abstract needs to be improved. Based on the way the authors represent their results, there are many variables which are significantly different between cold-water and thermal comfort groups. Also, compared to the sedentary control group, the magnitude of the difference is greater in the cold-water group than the thermal comfort group. However, the abstract does not reflect this aspect. Please revise the abstract appropriately.

Thank you very much for this remark. Abstract has been supplemented with a statement: “Moreover, compared to the sedentary control group, the improvement in energy metabolism was greater in the cold water group than in the thermal comfort group”.

  1. Line 17, the authors may want to add detailed information about the age and the number of animals.

Thank you very much for this remark. Abstract has been supplemented with a statement: “The study involved 32 male and 32 female rats aged 15 months, randomly assigned to: control sedentary animals, animals training in cold water at 5°C ± 2°C, animals training in water at thermal comfort temperature (36°C ± 2°C)”.

  1. Line 23, It is confusing to recognize the TAN as the adenylate nucleotide pool. The authors may want to come up with clearer abbreviations.

Thank you very much for this remark. Abstract has been supplemented with a statement: TAN (total adenine nucleotide – (ATP+ADP+AMP)).

  1. Introduction is lengthy. Some of the background needs to go either methods or discussion section. One example, the detailed measurement outcome information (line 96-101) may not be necessary.

Thank you very much for this comment, however we would prefer to leave this fragment in order to avoid the lack of explanation of the examined parameters.

  1. Figure legends, the author may want to modify # p level of significance compared to the 36°C group rather than 5°C group.

Thank you very much for this comment, however we would prefer to leave this because the comparison concerns the 5oC group

  1. Figures 5 and 6 seem to show the cross-section of the muscle fiber, but I cannot recognize the clear section in most panels. Is this due to the cutting, processing, staining, or imaging problem? The authors need to replace the figures, which show clear cross-section.

Thank you very much for this comment. According to Reviewer remark we corrected the immunohistochemistry figures.

  1. Figure 8 looks like a hand drawn illustration, which is not appropriate for the publication. Please utilize the standardized tools for the overall scheme.

Thank you very much for this comment. Overall scheme (Fig.10) has been removed.

Reviewer 3 Report (New Reviewer)

Comments and Suggestions for Authors

The article submitted by Bosiacki M. et al. investigates the potential beneficial impacts of cold-water exercise on mitochondrial biogenesis and energy status in muscles of aging rats.

In my opinion, the data presented by the Authors don’t provide relevant developments in the Sector. In any case, further research is required to learn more about the topic presented.

Overall, the manuscript is well structured, the literature is appropriate and recent, and the methodologies are accurate and detailed. Some typing errors are still present within the text, and I invite the Authors to make appropriate corrections.

However, the following points need to be addressed and clarified.

Major points:

1. The writing of the results needs to be completely revised as there are too many repetitions and few explanations. For example, the phrase "The rats underwent swimming training in either cold water at 5°C or thermoneutral water at 36°C for nine weeks. The control group of animals remained sedentary. During the first week of the study, the initial swimming session lasted for 2 minutes (on the first day) and was extended by 0.5 minutes daily until reaching a training time of 4 minutes (on the fifth day of the first week). The results are presented as means and standard deviations." is mentioned too many times and the Authors are asked to change it.

2. The graphs of all Figures need to be refined, making the values ​​on the y-axis comparable between male and female rats. The Figures must be divided into panels and named with letters.

3. The Authors need to test some genes/proteins in RT-qPCR or Western blot related to mitochondrial function and to the control of oxidative metabolism (NAMPT, FASN, ACACA, FGF21, UCP1 and MT-ND1). The Authors could measure NAD/NADH levels, NADK/G6PD activities and CD38/TRPM2 expression.

4. The AMPK/PGC1alpha/Sirt1 axis is known to control mitohormesis. The Authors are invited to also study AMPK and Sirt1.

5. Protein expression was calculated by ELISA in Figures 4, 7, and 8; in Figure 3 instead the Western blot technique was performed. Authors must make the experiments as uniform as possible. I ask the Authors to carry out the Western blot of Figures 4, 7 and 8 and the ELISA of Figure 3.

6. At what temperature are the rats in the control group kept? Authors must specify this in the manuscript.

Minor points

1. The images of the Western blots must be whole and not cut into small pieces. Please include different images in Figure 3.

2. The discussion is too long and needs to be shortened.    

3. The Authors have confused the Figure numbers within the manuscript. For example, Mfn2 is in Figure 7 and not in Figure 5 (lines 311-317). In line 376 they refer to Figure 8 and not to Figure 7.

4. The y-axis of panel I B of Figure 9 is wrong.

5. The Western blots of Mfn1, Mfn2, Opa1 and Drp1 are to be performed.

6. The diagram of the research procedure used in the experiment is in Figure 10 and not in Figure 7.

7. There is an error in the housekeeping protein used in the Western blot technique. In Figure 3, beta-actin is indicated while in the text it is GAPDH. Which is the right one?

8. The blue color of the font must be removed in the text.

9. Why was the 2^-ΔΔCt method used only for PGC1alpha? Furthermore, GAPDH primers are missing. This housekeeping gene must be specified in the materials and methods.

10. PGC1alpha is misspelled in lines 766 and 768. The sentences "Quantitative evaluation of the expression of PGC1A, Mfn1, Mfn2, Opa1, and Drp1 genes encoding mitofusin 1 (Mfn1), mitofusin 2 (Mfn2), optic atrophy 1 (Opa1), and dynamin-related protein 1 (Drp1) in the muscles was conducted using quantitative real-time polymerase chain reaction (qRT-PCR). " (lines 768/771)"The expression level of the PGC gene in each sample was compared with the endogenous control. " (lines 787/788) need to be rewritten correctly.

Author Response

Review 3

( )

(x)

( )

Comments and Suggestions for Authors

The article submitted by Bosiacki M. et al. investigates the potential beneficial impacts of cold-water exercise on mitochondrial biogenesis and energy status in muscles of aging rats.

In my opinion, the data presented by the Authors don’t provide relevant developments in the Sector. In any case, further research is required to learn more about the topic presented.

Overall, the manuscript is well structured, the literature is appropriate and recent, and the methodologies are accurate and detailed. Some typing errors are still present within the text, and I invite the Authors to make appropriate corrections.

However, the following points need to be addressed and clarified.

Major points:

  1. The writing of the results needs to be completely revised as there are too many repetitions and few explanations. For example, the phrase "The rats underwent swimming training in either cold water at 5°C or thermoneutral water at 36°C for nine weeks. The control group of animals remained sedentary. During the first week of the study, the initial swimming session lasted for 2 minutes (on the first day) and was extended by 0.5 minutes daily until reaching a training time of 4 minutes (on the fifth day of the first week). The results are presented as means and standard deviations." is mentioned too many times and the Authors are asked to change it.

Thank you very much for this remark. Captions under figures have been shortened throughout the manuscript according to Reviewer remark.

  1. The graphs of all Figures need to be refined, making the values ​​on the y-axis comparable between male and female rats. The Figures must be divided into panels and named with letters.

According to Reviewer remark all figures has been corrected making the values ​​on the y-axis comparable between male and female rats. The Figures must be divided into panels and named with letters.

  1. The Authors need to test some genes/proteins in RT-qPCR or Western blot related to mitochondrial function and to the control of oxidative metabolism (NAMPT, FASN, ACACA, FGF21, UCP1 andMT-ND1). The Authors could measure NAD/NADH levels, NADK/G6PD activities and CD38/TRPM2 expression.
  2. The AMPK/PGC1alpha/Sirt1 axis is known to control mitohormesis. The Authors are invited to also study AMPK and Sirt1.

Thank you very much for this reviewer's comment, unfortunately we do not currently have additional, extra-planned funds for the project and we cannot perform the proposed analyses. However, we greatly appreciate this comment and will definitely use it in future research.

  1. Protein expression was calculated by ELISA in Figures 4, 7, and 8; in Figure 3 instead the Western blot technique was performed. Authors must make the experiments as uniform as possible. I ask the Authors to carry out the Western blot of Figures 4, 7 and 8 and the ELISA of Figure 3.

Thank you very much for this reviewer's comment. Both techniques we use allow us to determine the amount of protein, of course the ELISA method is a quantitative method and the WB method is a semi-quantitative one. For this reason, we decided not to present the results using the WB technique and presented only ELISA results for all tested proteins.

  1. At what temperature are the rats in the control group kept? Authors must specify this in the manuscript.

Thank you very much for this reviewer’s comment. This information can be found in the Material and methods subsection. The animal facility maintained a constant temperature (23 ± 2 °C), air humidity (approximately 40%), and a 12-hour day and night cycle.

Minor points

  1. The images of the Western blots must be whole and not cut into small pieces. Please include different images in Figure 3.

The WB image was not prepared from small pieces, it is the entire WB image, we included the original WB in the correspondence to the editor. However, we decided not to present the results using the WB technique and presented only ELISA results for all tested proteins.

  1. The discussion is too long and needs to be shortened.    
  2. The Authors have confused the Figure numbers within the manuscript. For example, Mfn2 is in Figure 7 and not in Figure 5 (lines 311-317). In line 376 they refer to Figure 8 and not to Figure 7.

According to Reviewer's comment, all figures and their numbering have been checked and corrected.

  1. The y-axis of panel I B of Figure 9 is wrong.

Thank you very much for this remark. According to Reviewer's comment, all figures and their numbering have been checked and corrected.

  1. The Western blots of Mfn1, Mfn2, Opa1 and Drp1 are to be performed.

Thank you very much for this reviewer's comment. Both techniques we use allow us to determine the amount of protein, of course the ELISA method is a quantitative method and the WB method is a semi-quantitative one. For this reason, we decided not to present the results using the WB technique and presented only ELISA results for all tested proteins.

  1. The diagram of the research procedure used in the experiment is in Figure 10 and not in Figure 7.

Thank you very much for this remark. According to Reviewer's comment, all figures and their numbering have been checked and corrected.

  1. There is an error in the housekeeping protein used in the Western blot technique. In Figure 3, beta-actin is indicated while in the text it is GAPDH. Which is the right one?

Thank you very much for this comment, due to the semi-quantitative nature of the WB marking, we have removed it from the manuscript.

  1. The blue color of the font must be removed in the text.

According to the reviewer's comment we have marked only the changes currently corrected in blue, and we have removed all other blue markings.

  1. Why was the 2^-ΔΔCtmethod used only for PGC1alpha? Furthermore, GAPDH primers are missing. This housekeeping gene must be specified in the materials and methods.

We would like to thank the Reviewer for this comment. Of course, this omission has been corrected in accordance with the reviewer's comment

  1. PGC1alpha is misspelled in lines 766 and 768. The sentences "Quantitative evaluation of the expression of PGC1A, Mfn1, Mfn2, Opa1, and Drp1 genes encoding mitofusin 1 (Mfn1), mitofusin 2 (Mfn2), optic atrophy 1 (Opa1), and dynamin-related protein 1 (Drp1) in the muscles was conducted using quantitative real-time polymerase chain reaction (qRT-PCR). " (lines 768/771)"The expression level of the PGC gene in each sample was compared with the endogenous control. " (lines 787/788) need to be rewritten correctly.

We would like to thank the Reviewer for this comment. The sentence has been corrected in accordance with the Reviewer's comment.

Round 2

Reviewer 1 Report (New Reviewer)

Comments and Suggestions for Authors

The article is ready for publication after revision. 

Reviewer 3 Report (New Reviewer)

Comments and Suggestions for Authors

In light of the changes made by the Authors, I consider the manuscript ready for publication on IJMS.

This manuscript is a resubmission of an earlier submission. The following is a list of the peer review reports and author responses from that submission.

Round 1

Reviewer 1 Report

Comments and Suggestions for Authors

The authors present a study on the effects of cold water and thermoneutral swimming in a rat model. The results of this study indicate, through measurement of purine compounds and gene expression. that cold water swimming may have additional effects on mitochondrial biogenesis and dynamics,as compared to thermoneutral swimming. However, direct measurement of mitochondrial content and respiration is not presented. Overall this is a nicely designed study, but improvements could be made. Specific recommendations are listed below.

Make sure the format of gene symbols and protein symbols are correct throughout the paper: Gene symbols for rats should be italicized, with only the first letter in upper-case (e.g., Gfap). Protein symbols are not italicized, and all letters are in upper-case (e.g., GFAP).

Abstract:

Please reduce the length of the abstract it is 404 words long and the author guidelines indicate it should be 200 words.

Lines 22-24: "The rats were randomly assigned to four study groups: control sedentary animals, animals training in cold water at 5°C ± 2°C, animals training in water at thermal comfort temperature (36°C ± 2°C)." You mention 4 study groups, yet only three are listed.

Please define non-standard abbreviations used (i.e. Ado, TAN, AEC).

Introduction:

The first two paragraphs contain no references, however there are claims made about current thoughts and conclusions in the field. Please add appropriate references. In particular, please cite a source for the statement that the body's adaptive mechanisms are becoming less efficient leading to higher susceptibility to stress-inducing factors (line 54-55).

Results:

Was body weight measured? Were there any differences observed among the groups?

Did the authors measure mitochondrial content and/or respiration directly in these animals? If so, it would greatly benefit this paper.

Section 2.9 lines 299-303: "Additionally, the mRNA expression of Mfn2 in the muscles of female rats swimming in cold water was significantly higher (approximately 40%, p≤0.02) compared to those training in warm water. Furthermore, the mRNA expression of Mfn2 in the muscles of female rats swimming in cold water was also significantly higher (approximately 38%, p≤0.008) compared to those training in warm water (Fig. 4.II.B)." The last 2 sentences in this paragraph are repetitive, I believe the authors meant to reference a comparison of the warm water and control group for one of these sentences.

Although the authors mention higher ATP levels in males as compared to females in the swimming groups in section 2.1, sex comparisons are not mentioned after this. Did the authors compare the male and female groups for other endpoints besides ATP levels? Were any other differences found? If other comparisons were made and no male/female differences were found, it would be helpful to the readers to state this.

Discussion:

The discussion is long and wordy and almost reads like a review paper. It is recommended to shorten and focus it, making it more concise.

The authors state that there was an increase in the rate of energy metabolism and changes in mitochondrial biogenesis and dynamics. These conclusions are a bit of an overreach as these parameters were not directly measured. Rather, purine compound levels and gene expression was measured and these changes are inferred. The authors should rephrase the discussion to indicate that these parameters were likely changed, as indicated by ADP/ATP and gene expression. On the other hand, the paper would be strengthened if the authors have any data directly measuring mitochondrial respiration rates (via Seahorse or Oroboros) or amounts (could be assessed by measuring mitochondrial DNA or citrate synthase activity). Then the above statements could be validated. Without these direct measures, the authors should be clear that that these effects on mitochondria are being inferred based on gene expression and purine compounds.

Methods:

Please provide the basic details of the nutrient composition of the LSM and a source and catalog number if purchased from a company.

Please clarify if the control group rats were also placed in the empty glass tank for familiarization each day for 7 days at the beginning of the study prior to any training sessions, or if this was only done for the rats receiving swimming sessions.

The authors state that thigh skeletal muscle was used, please clarify which muscle groups were included of the thigh muscles.

Lines781-783: "The samples were subsequently kept at -80°C until further analysis of ATP, ADP, AMP, Ado concentrations, and the expression of PGC-1α mRNA and proteins (Mfn1, Mfn2, Opa1, Drp1)."  It may be helpful to reword this as It sounds like you measured the protein expression of Mfn1, Mfn2, Opa1, and Drp1, when you only measured the RNA expression of these genes.

The reference for the HPLC method [Smolenski et al. 1990] seems to not have been added to the reference list in the same format as other references. 

Author Response

Thank you very much for very thorough review. All the comments from the Reviewer have been addressed in the text of our manuscript.

Review 1

Comments and Suggestions for Authors

The authors present a study on the effects of cold water and thermoneutral swimming in a rat model. The results of this study indicate, through measurement of purine compounds and gene expression. that cold water swimming may have additional effects on mitochondrial biogenesis and dynamics,as compared to thermoneutral swimming. However, direct measurement of mitochondrial content and respiration is not presented. Overall this is a nicely designed study, but improvements could be made. Specific recommendations are listed below.

Make sure the format of gene symbols and protein symbols are correct throughout the paper: Gene symbols for rats should be italicized, with only the first letter in upper-case (e.g., Gfap). Protein symbols are not italicized, and all letters are in upper-case (e.g., GFAP).

Thank you very much for all the reviewer's comments, which we tried to complete as accurately as possible and to the best of our ability.

According to Reviewer remark We corrected gene and protein symbols.

Abstract:

Please reduce the length of the abstract it is 404 words long and the author guidelines indicate it should be 200 words.

According to Reviewer remark we corrected the length of the abstract.

Lines 22-24: "The rats were randomly assigned to four study groups: control sedentary animals, animals training in cold water at 5°C ± 2°C, animals training in water at thermal comfort temperature (36°C ± 2°C)." You mention 4 study groups, yet only three are listed.

According to Reviewer remark we corrected the number of control and study group.

Please define non-standard abbreviations used (i.e. Ado, TAN, AEC).

According to Reviewer remark we corrected  the abbreviation in abstract.

Introduction:

The first two paragraphs contain no references, however there are claims made about current thoughts and conclusions in the field. Please add appropriate references. In particular, please cite a source for the statement that the body's adaptive mechanisms are becoming less efficient leading to higher susceptibility to stress-inducing factors (line 54-55).

According to Reviewer remark we added references for the first two paragraphs of abstract.

Results:

Was body weight measured? Were there any differences observed among the groups?

The same model of study had been used by our research team previously:

  1. Lubkowska, A.; Dołęgowska, B.; SzyguÅ‚a, Z.; Bryczkowska, I.; StaÅ„czyk-Dunaj, M.; SaÅ‚ata, D.; Budkowska, M. Winter-swimming as a building-up body resistance factor inducing adaptive changes in the oxidant/antioxidant status. J. Clin. Lab. Investig. 2013, 73, 315–25.
  2. Lubkowska, A.; Bryczkowska, I.; Gutowska, I.; Rotter, I.; Marczuk, N.; Baranowska-Bosiacka, I.; Banfi, G. The effects of swimming training in cold water on antioxidant enzyme activity and lipid peroxidation in erythrocytes of male and female aged rats. J. Environ. Res. Public Health 2019, 16, 647, doi:10.3390/ijerph16040647.
  3. Bosiacki, M.; Gutowska, I.; Piotrowska, K.; Lubkowska, Concentrations of Ca, Mg, P, Prostaglandin E2 in Bones and Parathyroid Hormone; 1,25-dihydroxyvitamin D3; 17-β-estradiol; Testosterone and Somatotropin in Plasma of Aging Rats Subjected to Physical Training in Cold Water. Biomolecules. 2021, 21;11(5):616. doi: 10.3390/biom11050616.

In these papers, we described and discussed in detail the changes in body weight in the tested rats. Male rats swimming in low temperatures showed a decrease in body weight, which, as suggested by the authors of the study, was associated with mobilization of adipose tissue storage and increase energy expenditure. In contrast, when swimming at a thermally comfortable temperature, the decrease in body weight in males and females was due solely to the exercise. The authors also noted that the females that took part in swimming sessions at 5 °C at first had a decrease in body weight in the first two weeks of the experiment, then an increase until the end of sessions. Since the present study is a continuation of that research and was performed on available tissues from rats from the cited experiment, it may provide valuable data for the analysis of the effect of cold-water immersion on body mass according to Reviewer remark. 

Did the authors measure mitochondrial content and/or respiration directly in these animals? If so, it would greatly benefit this paper.

Thank you very much for this Reviewer comment. The present study is a continuation of the research and was performed on available tissues from rats from the cited experiment. Therefore, the tissue material that was used in the present study was previously frozen at -80oC by the time of analysis, hence it was not possible to isolate mitochondria and measure mitochondrial content and/or respiration directly. Such studies would undoubtedly be very valuable and would enrich the assessment of muscle energy metabolism in this animal model.

Section 2.9 lines 299-303: "Additionally, the mRNA expression of Mfn2 in the muscles of female rats swimming in cold water was significantly higher (approximately 40%, p≤0.02) compared to those training in warm water. Furthermore, the mRNA expression of Mfn2 in the muscles of female rats swimming in cold water was also significantly higher (approximately 38%, p≤0.008) compared to those training in warm water (Fig. 4.II.B)." The last 2 sentences in this paragraph are repetitive, I believe the authors meant to reference a comparison of the warm water and control group for one of these sentences.

Thank you very much for this Reviewer comment. According to Reviewer remark we corrected  the sentence.

Although the authors mention higher ATP levels in males as compared to females in the swimming groups in section 2.1, sex comparisons are not mentioned after this. Did the authors compare the male and female groups for other endpoints besides ATP levels? Were any other differences found? If other comparisons were made and no male/female differences were found, it would be helpful to the readers to state this.

Thank you very much for this Reviewer comment. Significant differences in the concentration occurred only in the case of ATP. We did not observe further statistically significant differences in the concentration of ADP, AMP, Ado (thus TAN and AEC) between female and male rats in groups of animals swimming in both warm and cold water. According to Reviewer remark we corrected Results section. 

Discussion:

The discussion is long and wordy and almost reads like a review paper. It is recommended to shorten and focus it, making it more concise.

The authors state that there was an increase in the rate of energy metabolism and changes in mitochondrial biogenesis and dynamics. These conclusions are a bit of an overreach as these parameters were not directly measured. Rather, purine compound levels and gene expression was measured and these changes are inferred. The authors should rephrase the discussion to indicate that these parameters were likely changed, as indicated by ADP/ATP and gene expression. On the other hand, the paper would be strengthened if the authors have any data directly measuring mitochondrial respiration rates (via Seahorse or Oroboros) or amounts (could be assessed by measuring mitochondrial DNA or citrate synthase activity). Then the above statements could be validated. Without these direct measures, the authors should be clear that that these effects on mitochondria are being inferred based on gene expression and purine compounds.

Thank you very much for this Reviewer comment. Unfortunately according to our best knowledge existing literature provides no data on the impact of swimming training in cold water on energy metabolism, mitochondrial biogenesis, and dynamics in aging skeletal muscles, making it difficult to compare our results with those of other authors. However according to Reviewer remark we corrected, shortened and specify  the discussion section. 

Methods:

Please provide the basic details of the nutrient composition of the LSM and a source and catalog number if purchased from a company.

During both the quarantine and experimental phases, the rats were provided with standard laboratory chow (Murigran pellet, Motycz, Poland) as their diet and had unrestricted access to tap water for drinking.

According to Reviewer remark we corrected Methods section. 

Please clarify if the control group rats were also placed in the empty glass tank for familiarization each day for 7 days at the beginning of the study prior to any training sessions, or if this was only done for the rats receiving swimming sessions.

The animals in the experiment underwent a series of swimming sessions over 9 weeks. Before the swimming sessions began, each rat from the control and study groups was introduced to an empty glass tank (dimensions: length 100 cm, width 50 cm, depth 50 cm) for 2 minutes to familiarize them with the training sessions and the experimental environment. This process was repeated for 7 days before the swimming sessions commenced.

According to Reviewer remark we corrected Methods section. 

The authors state that thigh skeletal muscle was used, please clarify which muscle groups were included of the thigh muscles.

 Thank you very much for this Reviewer comment. Rats' rat medial gastrocnemius muscle was collected during the animal dissection and immediately stored in liquid nitrogen.

According to Reviewer remark we corrected Methods section. 

Lines781-783: "The samples were subsequently kept at -80°C until further analysis of ATP, ADP, AMP, Ado concentrations, and the expression of PGC-1α mRNA and proteins (Mfn1, Mfn2, Opa1, Drp1)."  It may be helpful to reword this as It sounds like you measured the protein expression of Mfn1, Mfn2, Opa1, and Drp1, when you only measured the RNA expression of these genes.

According to Reviewer remark we corrected  the sentence.

The reference for the HPLC method [Smolenski et al. 1990] seems to not have been added to the reference list in the same format as other references. 

Reviewer 2 Report

Comments and Suggestions for Authors

In this study, Bosiacki et. al investigates the positive impacts of cold water exercise on mitochondrial biogenesis and muscle energy metabolism in aging rats. The research involved 64 aging rats, split into groups training in cold water, thermally comfortable water, and control sedentary conditions. Both cold and comfortable water exercises improved energy metabolism and mitochondrial dynamics, as indicated by enhanced metabolic rates and fusion-regulating protein expressions. The study suggests that cold water exercise can optimally adapt muscles to changing environmental conditions, potentially benefiting long-term bioenergetic muscle capacity. The study is nicely designed; however, it significantly lacks essential experiments to substantiate its findings. Following are my comments:

1.       The primary concern stems from the authors' reliance on mRNA levels of mitochondrial proteins like Opa1, Drp1, Mfn1, Mfn2, etc., for drawing conclusions. It's widely acknowledged that mRNA levels might not correlate with actual protein levels within a cell. To establish more solid conclusions, performing western blotting/immunolabeling for these proteins prior to drawing inferences would have been prudent.

2.       While the study deduced improved energy metabolism and mitochondrial dynamics through metabolite profiling and mRNA expression outcomes following cold water swimming, there is an absence of assessments regarding alterations in mitochondrial respiration or muscle tissue's mitochondrial structural changes.

3.       The discussion section exhibits a comprehensive literature review; however, the authors fall short in delivering a robust discussion and meaningful interpretation of their own study's data.

4.       Incorporating muscle regeneration assays could have been valuable in assessing the reversal of aging phenotype resulting from cold water swimming.

5.       The study overlooks the impact of cold-water swimming on autophagy, mitophagy, ROS production, mitochondrial membrane potential, and mitochondrial mass, which are crucial factors for comprehensive insight.

Author Response

Thank you very much for very thorough review. All the comments from the Reviewer have been addressed in the text of our manuscript.

Review 2

Comments and Suggestions for Authors

In this study, Bosiacki et. al investigates the positive impacts of cold water exercise on mitochondrial biogenesis and muscle energy metabolism in aging rats. The research involved 64 aging rats, split into groups training in cold water, thermally comfortable water, and control sedentary conditions. Both cold and comfortable water exercises improved energy metabolism and mitochondrial dynamics, as indicated by enhanced metabolic rates and fusion-regulating protein expressions. The study suggests that cold water exercise can optimally adapt muscles to changing environmental conditions, potentially benefiting long-term bioenergetic muscle capacity. The study is nicely designed; however, it significantly lacks essential experiments to substantiate its findings. Following are my comments:

  1. The primary concern stems from the authors' reliance on mRNA levels of mitochondrial proteins like Opa1, Drp1, Mfn1, Mfn2, etc., for drawing conclusions. It's widely acknowledged that mRNA levels might not correlate with actual protein levels within a cell. To establish more solid conclusions, performing western blotting/immunolabeling for these proteins prior to drawing inferences would have been prudent.

Thank you very much for this Reviewer remark. Of course, we realize that this is a serious limitation in our manuscript. Unfortunately, we currently do not have sufficient financial resources to purchase appropriate antibodies and perform WB analysis or immunohistochemical tests. We included this Reviewer remark in the limitation of the study.

  1. While the study deduced improved energy metabolism and mitochondrial dynamics through metabolite profiling and mRNA expression outcomes following cold water swimming, there is an absence of assessments regarding alterations in mitochondrial respiration or muscle tissue's mitochondrial structural changes.

The present study is a continuation of the research and was performed on available tissues from rats from the cited experiment.

1) Lubkowska, A.; Dołęgowska, B.; SzyguÅ‚a, Z.; Bryczkowska, I.; StaÅ„czyk-Dunaj, M.; SaÅ‚ata, D.; Budkowska, M. Winter-swimming as a building-up body resistance factor inducing adaptive changes in the oxidant/antioxidant status. Scand. J. Clin. Lab. Investig. 2013, 73, 315–25.

2) Lubkowska, A.; Bryczkowska, I.; Gutowska, I.; Rotter, I.; Marczuk, N.; Baranowska-Bosiacka, I.; Banfi, G. The effects of swimming training in cold water on antioxidant enzyme activity and lipid peroxidation in erythrocytes of male and female aged rats. Int. J. Environ. Res. Public Health 2019, 16, 647, doi:10.3390/ijerph16040647.

3) Bosiacki, M.; Gutowska, I.; Piotrowska, K.; Lubkowska, A. Concentrations of Ca, Mg, P, Prostaglandin E2 in Bones and Parathyroid Hormone; 1,25-dihydroxyvitamin D3; 17-β-estradiol; Testosterone and Somatotropin in Plasma of Aging Rats Subjected to Physical Training in Cold Water. Biomolecules. 2021, 21;11(5):616. doi: 10.3390/biom11050616.

Therefore, the tissue material that was used in the present study was previously frozen at -80oC by the time of analysis, hence it was not possible to isolate mitochondria and measure mitochondrial content and/or respiration directly. Such studies would undoubtedly be very valuable and would enrich the assessment of muscle energy metabolism in this animal model. We included this Reviewer remark in the limitation of the study.

  1. The discussion section exhibits a comprehensive literature review; however, the authors fall short in delivering a robust discussion and meaningful interpretation of their own study's data.

Thank you very much for this Reviewer remark. The research we have conducted is, to the best of our knowledge, the first on the impact of immersion in cold water on changes in mitochondrial metabolism, and energy production. Therefore, we could not directly present the results of other studies in the discussion and refer critically to our results. However, we have tried to shorten and improve the discussion in the manuscript and to interpret results.

  1. Incorporating muscle regeneration assays could have been valuable in assessing the reversal of aging phenotype resulting from cold water swimming.
  2. The study overlooks the impact of cold-water swimming on autophagy, mitophagy, ROS production, mitochondrial membrane potential, and mitochondrial mass, which are crucial factors for comprehensive insight.

Thank you very much for this Reviewer comment. Unfortunately, as we mentioned earlier, the research material that was used to perform these study had already been prepared by us and we used frozen tissues, hence it was not possible to isolate mitochondria and measure mitochondrial content or mitochondrial membrane potential. Muscle regeneration assays, mitochondrial membrane potential, and mitochondrial mass would require us to obtain new funding, organised a new project, and thus obtain new approvals for research on animals, purchase new animals, conduct a whole new experiment. We would love to conduct such wonderful experiments, but unfortunately, we currently do not have sufficient financial resources to perform new experiments. We included this Reviewer remark in the limitation of the study.

Limitation of the study

Our study of energy metabolism in the muscles of aging rats exposed to cold-water swimming training has provided strong evidence supporting improvements in muscle energy metabolism, biogenesis, and mitochondrial dynamics. These improvements were primarily observed at the mRNA expression level of the proteins examined. Extending our investigation to protein expression using western blotting/immunolabeling techniques would offer additional confirmation of heightened energy metabolism in aging rat muscle mitochondria. Additionally, a valuable direction for our upcoming research could involve exploring mitochondrial content, inner mitochondrial membrane potential, mitochondrial respiration, autophagy, mitophagy, and ROS production. The results from these investigations, combined with the measured concentrations of high-energy compounds and the expression of proteins that regulate mitochondrial dynamics in muscles, could serve as practical indicators for monitoring adaptive changes in aging muscles in response to physical exertion in cold water conditions.

Round 2

Reviewer 1 Report

Comments and Suggestions for Authors

I feel that the authors have addressed my concerns and made sufficient edits to the manuscript. It is now suitable for publication. I had a few minor recommendations that authors may want to take into account during the proof stage.

Recommendations:

1. To edit the abstract to briefly highlight the observed differences in the measured parameters between warm water swimming and cold water swimming. As it reads now there isn't anything mentioned about effects of the swimming temperature (only indication that both improve over sedentary controls), readers may pass by this manuscript on their search for differences induced by temperature if skimming the abstract.

2. Double check the formatting of gene and protein names per my previous recommendations. (Gene symbols for rats should be italicized, with only the first letter in upper-case (e.g., Gfap). Protein symbols are not italicized, and all letters are in upper-case (e.g., GFAP).) I see that authors fixed most of these correctly, but it looks like there were a few instances where not all formatting was applied (ie italics to genes, proteins not in all capital letters).

Comments on the Quality of English Language

Overall the English quality was good. There were a few typos I came across during my review, the authors may want to double check the manuscript during the proof stage.

Reviewer 2 Report

Comments and Suggestions for Authors

I want to express my appreciation for the effort authors put into addressing the concerns raised during the initial review of the manuscript titled “The effect of cold-water swimming on energy metabolism, dynamics, and mitochondrial biogenesis in the muscles of aging rats." I have carefully reviewed the revised manuscript and response letter.

I understand the challenges encountered by authors, such as funding shortages and sample availability, which can indeed pose significant hurdles in the research process. However, in the interest of maintaining the scientific quality and integrity of the journal, it is crucial that the issues highlighted in the initial review be adequately addressed in the revised manuscript. Regrettably, after reviewing the rebuttal and the revised manuscript, I find that the essential changes recommended during the initial review have not been satisfactorily incorporated into the manuscript. As such, I must maintain my previous decision to reject the manuscript for publication in its current form.

I want to emphasize that this decision is not a reflection of the importance of the research, or the challenges authors faced but rather a result of the manuscript not meeting the necessary standards for publication.